# The Utilization of Diffusion Tensor Imaging as an Image-Guided Tool in Brain Tumor Resection Surgery: A Systematic Review

**DOI:** 10.3390/cancers14102466

**Published:** 2022-05-17

**Authors:** Aiman Abdul Manan, Noorazrul Yahya, Zamzuri Idris, Hanani Abdul Manan

**Affiliations:** 1Functional Image Processing Laboratory, Department of Radiology, Universiti Kebangsaan Malaysia Medical Centre, Kuala Lumpur 56000, Malaysia; p97676@siswa.ukm.edu.my; 2Diagnostic Imaging and Radiotherapy, Faculty of Health Sciences, National University of Malaysia, Jalan Raja Muda Aziz, Kuala Lumpur 50300, Malaysia; azrulyahya@ukm.edu.my; 3Department of Neurosciences, School of Medical Sciences, Universiti Sains Malaysia, Penang 16150, Malaysia; zamzuri@usm.my; 4Department of Radiology and Intervensy, Hospital Pakar Kanak-Kanak (HPKK), Universiti Kebangsaan Malaysia, Jalan Yaakob Latiff, Kuala Lumpur 56000, Malaysia

**Keywords:** diffusion tensor imaging, preoperative planning, brain tumor, surgical approaches, systematic review

## Abstract

**Simple Summary:**

Diffusion tensor imaging (DTI) is an image-guided tool, especially in brain tumor resection surgery. Neuroimaging tools are essential for operative planning, particularly for maximizing tumor resection and, at the same time, preserving brain function. In this systematic review, we discuss the utilization of DTI in brain tumor resection, by looking into its ability to assess the perioperative approach, as well as evaluating its benefits for successful surgery. The present study proposes to use DTI as a vital neuroimaging tool for preoperative planning in brain tumor resection surgery.

**Abstract:**

The diffusion tensor imaging technique has been recognized as a neuroimaging tool for in vivo visualization of white matter tracts. However, DTI is not a routine procedure for preoperative planning for brain tumor resection. Our study aimed to systematically evaluate the effectiveness of DTI and the outcomes of surgery. The electronic databases, PubMed/MEDLINE and Scopus, were searched for relevant studies. Studies were systematically reviewed based on the application of DTI in pre-surgical planning, modification of operative planning, re-evaluation of preoperative DTI data intraoperatively, and the outcome of surgery decisions. Seventeen studies were selected based on the inclusion and exclusion criteria. Most studies agreed that preoperative planning using DTI improves postoperative neuro-deficits, giving a greater resection yield and shortening the surgery time. The results also indicate that the re-evaluation of preoperative DTI intraoperatively assists in a better visualization of white matter tract shifts. Seven studies also suggested that DTI modified the surgical decision of the initial surgical approach and the rate of the GTR in tumor resection surgery. The utilization of DTI may give essential information on white matter tract pathways, for a better surgical approach, and eventually reduce the risk of neurologic deficits after surgery.

## 1. Introduction

Preoperative planning in brain surgery plays a significant role in the successful outcome of the surgery, which eventually improves patients’ quality of life. Furthermore, preoperative planning in neurosurgery maximizes tumor resection, while preserving the brain’s neurological functions [1]. Diffusion tensor imaging (DTI) is one of the non-invasive neuroimaging tools for visualizing white matter tracts and pre-identifying tumor locations. This technique can also measure the interaction between the tumor and the surrounding areas of the brain, especially in the eloquent brain area [2]. However, this technique is not a routine technique for preoperative mapping, due to its artefacts and limitations, including the technique and preprocessing discrepancies that need to be addressed before data post-processing [3].

In previous studies, DTI was utilized in brain tumor patients, mainly to classify the tissue’s characteristics and understand the effects of tumor growth on the microstructural integrity of the surrounding brain tissue [4,5,6]. Besides, DTI was also used for navigating the anatomy of the tumor, giving essential routes for neurosurgery and to decide the surgical approaches [7,8]. Intensive research on DTI in brain tumor surgery has been carried out, mainly on the evaluation of brain tumor resection, and been incorporated with other modalities; for example, when combined with functional magnetic resonance imaging (fMRI). To date, no single technique has become the ultimate technique for understanding radiological assessment for brain tumor resection surgery [6]. Many methods have been developed and used alongside DTI, mainly to ensure efficacy, as DTI has been shown to have many limitations and artefacts [3,9,10]. Some studies even claimed that non-invasive DTI provided minor complementary information for brain mapping, which is critical in neurosurgery [11].

Thus, this systematic review investigates the effectiveness and benefits of utilizing DTI as a single neuro-navigation tool for brain tumor resection. The present review also compared DTI intervention group with non-DTI group. Finally, the present study evaluated the modification of the preoperative planning based on the DTI data findings, the re-evaluation of intraoperative DTI, and the best surgical approach.

## 2. Materials and Methods

### 2.1. Search Strategy and Study Selection

Two independent researchers conducted a systematic search using the National Centre for Biotechnology Information (PubMed) and Scopus electronic databases. The preferred reporting items for systematic reviews and meta-analyses guidelines (PRISMA) were used as the reporting guidelines [12] (Appendix A) and followed previous studies [13,14,15,16,17,18]. The search was performed to identify studies reporting diffusion tensor imaging, brain tumors, and preoperative mapping. We sought to evaluate clinical studies on brain tumor patients who had diffusion tensor imaging for brain resection surgery and a preoperative surgical planning review. This will aid in the re-evaluation of the surgical strategy, resulting in surgical decision modification and improved outcomes. We also included articles that made a comparison between the intervention of DTI and a control group. 

The article search was conducted between the earliest record and 27 January 2022. Search terms used included ‘brain neoplasm or brain cancer or brain lesion or brain tumour or brain tumor’ and ‘surgery approach or surgery operation or surgery planning or surgery process or surgical procedure’ and ‘DTI or diffusion tensor imaging or diffusion tensor tractography or tractography or fiber tracking or fibre tracking’; and the full advanced search used for each database can be found in Appendix A. We also manually searched the Google Scholar database for related papers in references and citations. There were no restrictions on the status of the publication or the date of publication. All records were collected into a final database after deleting duplicates, which were then screened by title and abstract. 

### 2.2. PICOS and Inclusion and Exclusion Criteria

Consensus for eligibility was reached through discussion and using the PICOS strategy, as summarized in Table 1. 

The inclusion and exclusion criteria were also screened. The papers were limited to adult humans aged >17 years old, and the articles were written in English. No limitation was set for sample size and year of publication. Second, we excluded articles that also utilized other imaging modalities, such as computed tomography (CT), positron emission tomography (PET), ultrasound, navigated transcranial magnetic stimulation (nTMS), fMRI, and other diffusion-weighted techniques, i.e., high angular resolution diffusion imaging (HARDI). However, magnatice resoncance imaging (MRI) were included. Review papers, systematic review papers, and technical notes were also excluded. 

## 3. Results

### 3.1. Data Extraction and Study Design

Seventeen studies fulfilled all the criteria for the systematic review, out of 2270 publications. The detailed flow processes of article selection for this systematic review are presented in Figure 1.

Based on the quality assessment tools published by the National Heart, Lung, and Blood Institute, these studies were assessed as reasonable and fair (Appendix A). This systematic review was registered under the International Prospective Register of Systematic Reviews (PROSPERO)—CRD42022314014. The articles were published from the year 2005 to 2021. The studies included both prospective studies (*n* = 9) [7,8,19,20,21,22,23,24,25] and retrospective (*n* = 8) [1,26,27,28,29,30,31,32]. Data collection was performed by extracting information on demographic data, composed of the author, year of publication, type of study, origin country, number of patients and participants, patients’ mean age, tumor type and its histology, tumor location, and the details of the control group, as shown in Table 2. For the main objective, we evaluated, extracted, and tabulated the information in Table 3 and Table 4. The information comprised (1) the evaluation of white matter tract tractography of interest in preoperative planning and intraoperative assessment; (2) type of surgery, any modification of the surgical approach planned based on DTI tractography assessment; (3) surgical outcome of the study; and, finally, a (4) summary of the main findings of the selected articles, as well as an overview of the (5) comparison between the DTI intervention group and the control group, are tabulated.

### 3.2. Participants

The sample size ranged from 8 to 230 participants, including controlled participants. The total number of patients reviewed was 877, with 234 control patients, with slightly more male patient recruitment (*n* = 360) compared to female patient recruitment (*n* = 283). The age range for these studies was 1 to 87 years old, and we included five studies [21,24,26,27,30] that had pediatric patients, but this would not affect our objective, as only a few patients were involved. All types of tumor were studied. Although there were studies with patients with cavernous malformation brain lesions, our study excluded only the irrelevant patients, who were diagnosed with arteriovenous malformations, cavernous angiomas, and cavernous malformations, as the authors listed all of the individuals’ tumor types. Thus, we included these two papers [19,26]. The tumor site was not exclusive; however, three of the publications discussed tumors exclusively in the brainstem [1,26,30]. We decided to include all tumors from all locations. This information is summarized in Table 2. The white matter tracts of interest in the studies were primarily those in close proximity to the tumor and those associated with brain function, such as the corticospinal tract related to motor function [19,20,27,30,31,32], optic radiation responsible for eyesight [1,7,8,21,23,29,31], and inferior longitudinal fasciculus and arcuate fascicles that link to language [7,8,20,22,23,28,29,31]. These tracts were relevant for postoperative deficit assessment. In the following section, we will discuss the use of DTI in brain resection surgery.

### 3.3. Utilization of DTI in Brain Resection Surgery

All of the studies discussed the utilization of the DTI in brain tumor resection. One of them was primarily concerned with the ability to reconstruct the fiber tract in the presence of a brain tumor [1,7,8,19,20,21,22,23,24,25,26,27,28,29,30,31,32]. In addition, the use of DTI could predetermine the preplanning surgical approach [8]. Evaluation of DTI was performed in comparison with control patients, non-DTI, and DTI patients’ groups [1,21,22,23,25,28]; intraoperatively evaluation [1,7,8,19,20,21,22,23,24,25,26,27,28,29,30,31]; and quantified the sensitivity and specificity of the DTI technique in its tract identification, tumor resection rate, and mobility prediction [20,21,23,24,27,31]. We emphasized all these benefits of DTI for our primary investigation on whether re-evaluation and modifications of the preplanning surgery were made [7,8,26,27,29,30,32]. Finally, the effectiveness of the surgery was measured by positive indications of the outcome, mainly the gross total resection of the tumor [1,7,8,22,23,25,26,27,29,32] and any development of postoperative neurological deficits [1,19,20,21,23,24,26,28,30,31,32].

### 3.4. Comparison to the Control Group (n = 6)

We compared the DTI group and the non-DTI group for evidence to prove that the DTI is a practical neuro-navigation tool for neurosurgery. Most of the studies conducted [1,21,22,23,25,28] compared the DTI result with the outcome of the surgery. The investigations contrasted the DTI and non-DTI groups, with the DTI group having a higher chance of total gross resection and a lower risk of postoperative impairment [1,22,25]. Zakaria et al. looked into automated whole-brain tractography (AWBT) and compared it to brain mapping in groups with and without it. Although the risk was similar in both groups, those with AWBT with brain mapping were more likely to recover from any postoperative neurological abnormalities [28].

In a preliminary study on the clinical application of DTI on the optic pathway by Hajiabadi et al. (2016), comparisons were made of the assessment of preoperative, intraoperative, and postoperative effects on visual impairment caused by the compression of the optic chiasm. The control group recruited were patients diagnosed with other diseases. They claimed that patients with an abnormal visual impairment score had optic chiasm compression. For the control group, the visual examinations recorded normal visual status in preoperative and postoperative periods, revealing that the fibers crossing the optic chiasm were without any alterations [21]. In addition to greater GTR in high-grade glioma (HGG) patients and better postoperative Karnofsky Performance Status score ratings, DTI navigational surgery gave a 43 percent reduction in the risk of death during neurosurgery compared to non-DTI navigational surgery [25].

### 3.5. Preoperative and Intraoperative DTI Tractography Evaluation (n = 4)

Nimsky et al. (2005), Hajiabadi et al. (2006), Aibar-Duran et al. (2020), and Maesawa et al. (2010) investigated preoperative tractography and intraoperative tractography [21,23,24,27]. All of these studies reported the use of intraoperative scanned DTI and its evaluation compared to preoperative DTI, except for Aibar-Duran et al. who evaluated the intraoperative navigated DTI. Nimsky et al. (2005) and Maesawa et al. (2010) documented fiber shifting intraoperatively, and the inward and outward shifting were determined to be undetectable preoperatively [24,27]. According to Hajiabadi et al., the postoperative mean distance between the optic tracts in tractography decreased intraoperatively. In the majority of the patients, there were no crossing fibers in the optic chiasm prior to surgery. Intraoperatively, however, the fibers were detected in five more individuals [21]. A study conducted by Aibar-Duran et al. (2020), which compared intraoperative navigated DTI with a control group, concluded that intraoperative navigated DTI tractography would shorten the awake surgery time [23].

### 3.6. Surgical Approach or Preoperative Plan Modification Concerning DTI Data or Tractography (n = 7)

The assessment of fiber trajectories of the tumor patient preoperatively or intraoperatively can result in the modification of initial surgical planning. Seven studies discussed these changes. Romano et al. (2009) delineated that the preoperative DTI depicted an inaccurate shifting of the central white matter tract, and that revision of surgical planning was needed to determine the resection margin. The study concluded that MR-tractography modified the planned surgical procedure in nearly 82 percent of cases [7]. Similarly to this, the location of the surgical approach was changed in Cao et al. (2010), and this modification only happened in one out of eight patients for the brainstem lesion [26]. Maesawa et al. (2010) outlined conditions to determine the surgical approach regarding the need for intraoperative DTI rescanning, resulting in total resection of the brain tumors of patients [27].

Research conducted by Faust and Vajkoczy (2016) revealed not the modification of surgical planning, but the predetermined surgical approach based on the location of the tractography of optic radiation tracts in relation to the tumor. Temporal tumor growth effects on optic radiation caused fixed pattern displacement, and these patterns provided a potential entry point for surgical tumor resection [8].

In contrast to these real studies, Alexopoulous et al. (2019) performed a stimulation of neurosurgeons’ decision-making in preoperative surgical planning. Two blinded neurosurgeons analyzed preoperative DTI and MRI scan imaging and determined the surgical decision for the brain tumor resection. However, after analyzing DTI tractography data on a few of their patients, they discovered that this technique had no effect on the surgical strategy. Nonetheless, it did corroborate the resection decision to move from total resection to subtotal resection, initially solely based on MRI, resulting in better surgical results [29]. Similarly, Buchmann et al. observed that DTI fiber tracking did not influence surgical planning or intraoperative course. However, post hoc imaging DTI offered the neurosurgeon the opportunity to adjust the surgical approach strategy in one of the instances analyzed [32]. In contrast to Alexopoulous et al. and Buchmann et al., Xiao et al. considered a change in surgical approach in more than a quarter of patients, based on the DTI data results in their study on the visualization of corticospinal tract. Following the DTI data, surgical approach selection was more diversified and particular than prior to the DTI result, with a preference for far lateral approaches [30].

### 3.7. Evaluation of DTI Tractography Utilization on Postoperative Surgical Outcome

The surgical outcome was assessed to evaluate GTR and the patients’ postoperative neurological function. In the present study, several of the selected papers reported their achievements based on the increasing decision rate in deciding the total gross resection of the patients [1,7,8,22,23,25,26,27,29], and the neurological deficits improved after time postoperatively [1,19,20,21,23,24,26,28,30,31,32], in the case where DTI was integrated as a preoperative planning tool. Many of the studies concluded that with the aid of DTI, the modification to initial preoperative planning could have been made mainly to exchange the outcome, in terms of maximizing the removal of the tumor [1,7,8,19,20,21,22,23,24,25,26,27,28,29,30,31,32]. In addition, the optimum surgical decision for tumor resection was given to the patients, mainly in cooperation with determining the best surgical type and approach [8,25] or changing the initial preoperative planning intraoperatively [7,26]. The utilization of DTI for postoperative outcome improvement mainly resulted in the improvement of neurological deficits. Some studies agreed that this could predict postoperative deficits and functional preservation [31].

## 4. Discussion

The present study discussed the utilization of DTI in brain tumor preoperative planning, to evaluate the benefits of DTI and its significance for successful neurosurgery. We found a specific selection of articles that used only DTI neuroimaging for preoperative, intraoperative, and postoperative assessment, aside from routine MRI scanning. These selected studies highlighted a few points. First of all, DTI is the only tool for in vivo visualization of white matter tract pathways, and it may aid in understanding the anatomy, architecture, and microstructure of the tracts, as well as localizing tumors involving the fiber tracts [1,7,8,19,20,21,22,23,24,25,26,27,28,29,30,31,32]. The sensitivity and specificity were proven by identifying tracts as 95% and 100%, respectively [33]. Most of the tracts were successfully reconstructed [1,7,24]; however, the technique has improved over time. The DTI technique is one of many tools used for preoperative planning and its impact on neurosurgical outcomes could depend on multiple factors. However, from our review, using DTI preoperatively could predict potential postoperative neurological deficits and increase the possibility of preserving brain functions [1,19,20,21,22,23,24,26,27,28,29,30,31]. Therefore, the introduction of DTI for preoperative planning and intraoperatively would be a significant step for a better surgical outcomes, based on the comparisons made in this study [1,21,22,23,25,28].

We compared the interventions in DTI and control groups, to assess their efficacy. This comparison showed that the most DTI utilizing group had better surgical decision outcomes on total gross resection. Furthermore, the postoperative neurological deficits were mostly improved and required less time for neurosurgery compared to the control group [1,21,22,23,25,28]. This is parallel with the statement by Aibar-Duran et al. 2020, in which the sensitivity and specificity for predicting complete tumor resection were 88% and 62.5 for the non-DTI group and 100% and 80% for the DTI group [23]. Thus, DTI provided the optimum surgical decision-making in deciding GTR and revealed the surgical path that could potentially alleviate the patient’s neurological deficits postoperatively.

Preoperative tractography was utilized for surgical planning, so that a proper plan could be executed. Robust analysis of the white matter tract tractography pattern, colors, and fiber pathways would benefit a neurosurgeon, as additional information is essential. The characterization and criteria of white matter tract microstructure interaction with the different types of the tumor would give an initial suggestion on how the surgical approach should be assigned [1,10]. In addition, by also analyzing the tumor location and its shifting effect on the white matter tract, this minor piece of information could become the permanent method for surgical approaches, as Faust and Vajkoczy (2016) reported. The surgical procedure was predetermined by the optic radiation fiber shift pattern, which could benefit the cases where DTI is unavailable [8].

In this review, we analyzed all types of tumors and grouped them into HGG, low-grade gliomas (LGG), and metastasis. Different types of tumors may have different interactions and effects on the fiber tract, which may necessitate surgical intervention [1,4]. Unlike HGG and metastasis, most LGGs have a higher chance of surgical modification [34]. Witwer et al. (2002) reported four types of characterization of white matter tract microstructure changes: displacement, infiltration, edema, and destruction. The classification of these characteristics depends on the value of fractional anisotropy and the visualization of the normality of the microstructure orientation of the white matter tract [4]. Papers have discussed the classification of fiber tracts in different tumors, such as benign and malignant tumors, and showed that most of the malignant tumors depicted a destruction pattern [35,36,37]. In terms of the rate of tumor resection, infiltrating tumors could only yield a subtotal resection, and HGG usually ended with a total resection [34,36]. We have included supratentorial tumors and a few papers on brainstem tumors. It should be known that the treatment and the prognosis of tumors in the brainstem are different, and, as proposed by Cao et al., preplanning of brainstem tumor surgery should be done with an individualized approach [26,35]. This should be addressed properly, as we wanted to conduct a systematic review of the usefulness of DTI in tumor resection surgery in a broader sense.

Intraoperatively navigated tractography gives a more precise fiber tracking depiction than preoperative tractography, and it has the advantage of shortening the surgery time [23]. In addition, brain shifting of the fiber could happen in the brain, and the location of the fibers visualized in preoperative DTI might be different from the intraoperative DTI [24]. White matter tract shifting occurs for many reasons, including gravity or head positioning, surgical equipment, tissue loss, tissue fluid, and tumor type. In the intraoperative session, the brain shifting would be predictable in the case of the head position during the operation. However, the inward and outward shift are unpredictable and mutually exclusive [24,38].

Intraoperative DTI assessment contributes to successful awake-surgery, plus any preoperative error can be corrected intraoperatively [20,33]. Although the utilization of DTI is sufficient, the establishment of brain-mapping for preoperative planning is needed [28]. Thus, it is crucial to integrate this with other navigation methods, such as DES or direct cortical electrostimulation, which is the gold standard for brain tumor resection methods in awake surgery, as an indication of the functional border of the brain [19,20,38,39,40]; as well as the use of mixed techniques, which include fMRI for a complete understanding of the functional part of the brain [38]. This is needed due to the finding that although the predictive accuracy of the DTI technique to locate the tracts was undeniably high, the specificity prediction of the tract correlated with the function and stimulation was low intraoperatively [31].

As shown by Hajiabadi et al. and Voets et al., DTI fiber tracking could predict the postoperative deficit [21,31]. A study by Sollmann et al. (2016) showed that preoperative DTI fiber tracking derived from navigated transcranial magnetic stimulation detected the hemispeheric connectivity, which could contribute to surgery-related aphasia [41]. Postoperative neurological deficits could be avoided with proper surgical planning, and from our standpoint, DTI tractography could assist in preserving the eloquent functioning of the brain, mainly when used together with intraoperative subcortical stimulation combined with neurophysiological assessment [33].

The act of modification and re-evaluation of surgical plans indicates more information is required, and the utilization of DTI should have been done routinely. Most studies had to re-evaluate the planned surgical process in the intraoperative courses [7,27]. Based on our evaluation, the modification was mainly done on the initial GTR, whether the resection was a subtotal or total removal. Ultimately, as backed by most studies, GTR of gliomas, either HGG or LGG, would increase the median survival rate by more than 120% [42]. While most agreed that this only concerns the GTR changes, some have shown its effect in modifying the location of surgical approaches [8,42].

DTI tractography, as an essential tool in neurosurgery care and treatment, still raises questions, as there is no reliable standard [43]. Some findings argue that preoperative DTI is simply one tool of many, and that it does not give any positive feedback in neurosurgery as it cannot influence surgical strategy or modify surgical planning. In this review, Buchmann et al. and Alexopoulos et al. gave us two sides of the same coin [29,32]. A recent systematic review and meta-analysis study shared their analysis, suggesting either a combination of DTI, fMRI, and intraoperative MRI used in intraoperative neuronavigation, or by themselves, was insufficient to conclude that these advanced imaging techniques have the potential to influence the GTR and neurological preservation [44]. The use of imaging modalities in surgery could increase the GTR rate; however, this would not improve the surgical outcome. In their analysis of iMRI studies, the authors pointed out that the imaging modality could permit better GTR and preserve the eloquent area in incomplete resection, when GTR not feasible. However, it is important to highlight that, in regards to their study on DTI, DTI has not been extensively studied and an exclusive meta-analysis on DTI studies should be done [44].

### Limitations

We should point out some limitations of our studies, by recognizing the small sample sizes of the selected studies. The DTI and tractography analysis processes have some limitations. The standardization of the fractional anisotropy and angle threshold could be hard to achieve due to edema and infiltration surrounding the tumor [23]. The cons of DTI are that it is user-dependent and requires a vast amount of anatomical knowledge. Importantly, DTI analysis requires understanding the physical nature of fiber, which includes crossing fibers, brain shifting, and corresponding parameters. These are a few examples of issues identified, from the many more addressed by the studies [10,42,45], and these could be systematically reviewed. Understanding the technical aspects of DTI could give us a better view of our study, as subtopic on the DTI acquisition parameters should be done. A magnetic resonance with a field strength of 1.5 Tesla is sufficient for diagnostic purposes when properly protocoled; nevertheless, 3 Tesla is regarded as the ideal instrument for DTI [46]. Quantitative DTI, fractional anisotropy, and mean diffusivity could provide possible indicators for successful surgery. Unlike other diseases, such as epilepsy or multiple sclerosis, the primary goal of DTI in neurosurgery is to visualize the fiber tracts, without delving deeper into its quantitative impact, which may be beneficial in tumor resection procedures and for postoperative neurological preservation. Combining more modalities would benefit the utilization of DTI in understanding the brain structure in neuronavigation. It is sufficient for the eloquent part of the brain to see the behaviors that cater to the brain’s functional part. The main reason why most preoperative DTI uses awake surgery is to ensure the validation of functional parts of the brain; mainly to preserve them. From the historical timeline of the progression of DTI, the utilization of the technique could be seen as a slow process. There are many other techniques of tractography, and DTI tractography data is the most utilized in the clinical field. Diffusion-weighted MRI provided the sequences for tractography; however, a thorough understanding in the processing algorithm is needed [43]. With the advance of scientific research and development, the newest mathematical models of diffusion MRI techniques, which is HARDI, and diffusion spectrum imaging give alternatives to avoid the shortcomings of DTI tractography [47]. In future studies, the emphasis will be on the use of advanced techniques such as HARDI. There needs to be a quantitative evaluation of its effectiveness. A meta-analysis could be the next review, to see if there are significant quantitative findings.

## 5. Conclusions

We have reviewed some of the benefits of utilizing DTI in neurosurgery, ensuring that many more will be discovered in the future. There are pros and cons to using DTI, mainly regarding its utilization as a single technique. Past and ongoing studies on DTI have shown the effects of DTI’s usage perioperatively and postoperatively. It can give essential information on the related white matter tract pathways and their relation to the tumors’ features, for a better surgical approach; and to a certain extent, potentially decreases the neurological deficit in patients’ quality of life. We can conclude that preoperative DTI has given us important information on fibers, and eventually, could help in evaluating operative planning, based on the evidence when compared to the non-DTI group. Intraoperative DTI, as well as intraoperative navigated DTI, should be used, and validation of the functional white matter tract would be possible with the aid of intraoperative subcortical stimulation and neurophysiological assessment, as collecting as much information as possible about the surrounding tissue of the tumor could assist in alleviating the postoperative deficit. However, more studies are needed, and applications need to be performed and validated, mainly on the use of DTI quantitative parameters, to help with neurosurgery resection. So that artefacts and complications may be controlled and managed in the future, the DTI technique should be used as a routine neuroimaging procedure, mainly in any neurosurgery and diseases that could potentially damage the white matter tract fibers.

## Figures and Tables

**Figure 1 cancers-14-02466-f001:**
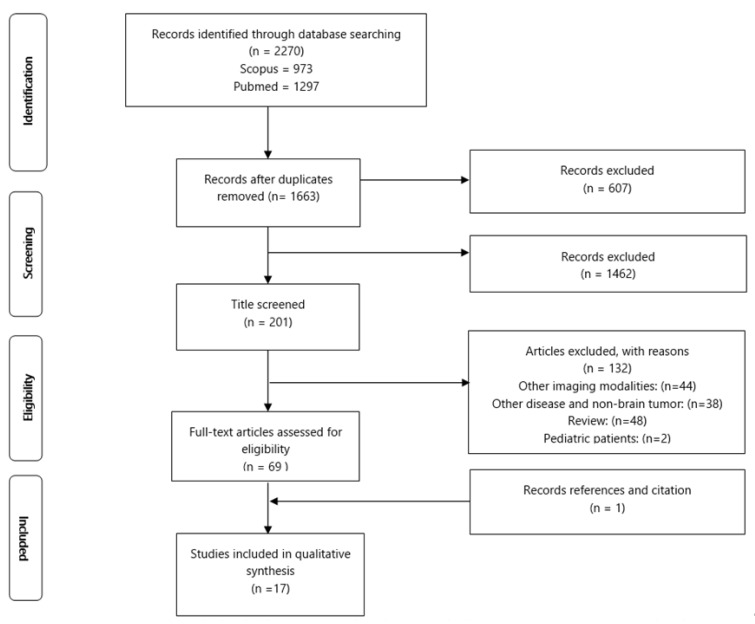
Flow diagram of the PRISMA study selection process.

**Table 1 cancers-14-02466-t001:** PICOS strategy for selection of the study.

PICOS	Criteria
P—Patients	Adult brain tumor patients.
I—Intervention	Underwent DTI scanning for preoperative planning.
C—Comparison	Re-evaluation of preoperative DTI and intraoperative DTI, modification of the preoperative planning based on DTI data, comparison with the non-DTI control group.
O—Outcome	Surgical decision and outcome.
S—Study	Only original clinical studies were selected.

**Table 2 cancers-14-02466-t002:** Demographic data of the patients, control participants, and tumor characteristics.

No	Author (Year): Country [Ref]	No of Patients (Male/Female)	Mean Age, (Age Range)	Tumor Type	Tumor Location	Control Participant
Prospective Study						
1	Okada et al. (2006): Japan [19]	8 (4/4)	41 years, (23–58)	Intracranial space-occupying lesions; Included HGG	Frontal, parietal lobe, pons	NR
2	Nimsky et al. (2005): Germany [24]	37 (20/17)	45.2 ± 21 years, (6–77)	Supratentorial gliomas; Included HGG and LGG	NR	NR
3	Wu et al. (2007): China [25]	118 (40/78)	Patients 40.8 ± 15.6 years, (6–75) Control 38.0 ± 16.5 years, (6–70)	Cerebral gliomas; Included HGG and LGG	Frontal, temporal, pariental, insular, central, occipital, basal ganglia, thalamus	120 (78/42) Control patients underwent craniotomies using 3-D navigational MRI only
4	Romano et al. (2009): Italy [7]	28 (19/9)	(38–77)	Intra-axial cerebral tumor; Included HGG, LGG and Metastasis tumor	Thalamus, fronto-parietal, frontal, parietal, temporal, temporooccipital	NR
5	Bello et al. (2010): Italy [20]	230	NR	Gliomas Included HGG and LGG	Precentral, Rolandic, parietal, temporal, insula	NR
6	Hajiabadi et al. (2015): Germany [21]	25 (15/10)	53.08 ± 18.61 years, (11–87)	Suprasellar mass lesion; Included HGG, LGG	Hypothalamus-pituitary	6 control patients with normal vision
7	Faust and Vajkoczy (2016): Germany [8]	113 (70/43)	54 ± 16 years	Intraaxial tumor; Included HGG, LGG and Metastasis tumor	Temporal	NR
8	Zhang et al. (2020): China [22]	21(13/8)	Patients 53.29 years Control 48.24 years	Intracranial tumor: Included HGG, LGG, and Metastasis tumor	Frontal, precentral gyrus, temporal, cerebral falx	21 (11/10) control patients underwent preoperative MRI only
9	Aibar-Duran et al. (2020): Spain [23]	37 (25/12)	53.8 years, (33–75)	Brain tumor in eloquent areas; Included HGG, LGG, and Metastasis tumor	Temporal or insular, frontal, parietal	18 control patients with no intraoperative navigated DTI
**Retrospective Study**						
10	Yu et al. (2005): China [1]	16 (12/4)	Patients 51.7 years, (20–72) Control 52.5 years (25–68)	Cerebral tumor; Included HGG, LGG and metastasis tumor	Brainstem	24 (17/7) control patients’ MRI data with suspicion of involvement of the pyramidal tract
11	Cao et al. (2010): China [26]	9 (5/4)	30.1 years, (4–49)	Brainstem lesion; Included HGG, LGG	Brainstem, (pons, medulla oblongata, midbrain)	NR
12	Maesawa et al. (2010): Japan [27]	28 (17/11)	46.5 years, (13–68)	Intracranial tumor; Included HGG, LGG	Deep-seated tumor located 20 mm of CST	NR
13	Buchmann et al. (2011) Germany [32]	19 (13/6)	49 years, (16–72)	Intracranial tumor; Included HGG, LGG, Metastasis tumor	Frontodorsal, frontal, precentral, insular, temporomesial, central, parietal, cingular	NR
14	Zakaria et al. (2017): USA [28]	28 (17/11)	Patients 51.75 ± 17.78 years, (26–76) Control 56.11 ± 11.23 years, (27–86)	Brain tumor within eloquent areas; Included HGG, LGG, and Metastasis tumor	Parietal, frontal, temporal, frontal-parietal, frontal-temporal	45 (30/15) control patients with non-mapping preoperative planning
15	Alexopoulos et al. (2019): USA [29]	15 (11/4)	58.3 years, (45.5–71.5)	Supratentorial tumor	Frontal, parietal, temporal, occipital	NR
16	Xiao et al. (2021) China [30]	54 (31/23)	17.6 years, (1.9–62.2)	Brainstem glioma; Included HGG, LGG	Brainstem	NR
17	Voets et al. (2021) UK [31]	91 (48/43)	49.2 years, (19–74)	Intrinsic Brain tumor; Included HGG, LGG and Metastasis tumor	NR	NR

Abbreviations: NR: not reported, MRI: magnetic resonance imaging, HGG: high-grade gliomas, LGG: low-grade gliomas, CST: Corticospinal tract.

**Table 3 cancers-14-02466-t003:** Comparison between the assessed DTI group and the control group.

No	Author	Main Findings
1	Yu et al. (2005) [1]	DTI group gave a better GTR outcome and less postoperative deficit in comparison to the control group
2	Wu et al. (2007) [25]	DTI navigational gave a better GTR in HGG than LGG and a higher KPS score and represented a 43.0% reduction in death risk compared to control
3	Hajiabadi et al. (2015) [21]	VIS on assessed DTI group was reduced from the compression of optic chiasm, as compared to control which have normal structure of the fiber optic chiasm
4	Zakaria et al. (2017) [28]	DTI brain mapping group’s postoperative neurology deficits improved in comparison to control
5	Aibar-Duran et al. (2020) [23]	Intraoperative navigated tractography group had more complete resection, less postoperative neurological damage, and shorter surgery time than the control group
6	Zhang et al. (2020) [22]	The postoperative KPS score in the DTI group was significantly better than the control group, although there are no significant difference in GTR between the two groups

Abbreviation: DTI; Diffusion tensor imaging GTR; Gross Total Resection, HGG;high-grade glio-mas, LGG; low-grade gliomas, KPS; Karnofsky Performance Scale, VIS; Visual Impairment score.

**Table 4 cancers-14-02466-t004:** Surgical Planning Approach Modification, Preoperative and Intraoperative Assessment Based on DTI and Tractography Data and Its Surgical Outcome.

Author (Year)	White Matter Tract of Interest	Assessment of White Matter Tract (WMT) during		Type of Surgery	Modification of Surgical Approach or Plan by DTI Tractography (Yes /No)	Surgery Outcome	Main Finding
		Preoperative	Intraoperative			GTR/Postoperative Deficits Assessment	
Yu et al. (2005) [1]	Pyramidal, Corpus Callosum, Optic Radiation	Preoperative depiction of DTI and WMT characterization evaluation pre-determined surgery approach.	NR	Craniotomy	No	GTR: DTT group patients were higher, compared to the control. Postoperative deficit: locomotive function of the DTT group was improved.	The GTR and surgical approaches were determined by the type of WMT characterization depicted by DTT.
Okada et al. (2006) [19]	CST	Preoperative DTI of WMT depicted for surgical planning	DTI tractography used with MEP	Craniotomy	No	No postoperative neurological deficits.	Affective combinations of DTI tractography with MEP.
Wu et al. (2007) [25]	Pyramidal Tracts	Preoperative DTI and MRI were used and compared to only MRI scan control.	NR	Craniotomy	No	GTR: Higher chance of HGG in DTI group. Postoperative deficits: KRS score higher in DTI group.	DTI navigational neurosurgery gave reduction in death risk compared to the control group.
Romano et al. (2009) [7]	Pyramidal tract, Optic Radiation, Arcuate fasciculus	Preoperative DTI of WMT depicted for surgical planning	Assessment trajectories of fibers, some needed for repeated tractography.	Craniotomy, Corticotomy	Yes, modification of resection margin and surgical approach.	GTR: 64% successful predefined on resection margin, allowed further resection. Postoperative deficits: improved with successful DTI trajectories.	The MR DTI altered preoperational planning and modified the surgical approach to craniotomy in 21% of the patients.
Nimsky et al. (2005) [24]	Pyramidal tract, corpus callosum	Preoperative DTI depicted WMT fiber in the vicinity of the tract in error less than 20 mm.	Intraoperative DTI marked inward or outward shifted range of WMT	Craniotomy	No	Postoperative deficits: only one patient encountered new neurological deficits.	Fiber shifts were evaluated by intraoperative DTI, resulting in a shifting pattern inward or outward of WM fibers.
Cao et al. (2010) [26]	CST, medial lemnisci	Preoperative DTI tractography used for individualized surgical approach.	One out of eight patients needed to evaluate the DTI tractography.	Craniotomy	Yes, from suboccipital to restomastodial approach.	GTR: total resection was achieved in four patients. Postoperative deficits: neurological examination improved.	MRI scans were sufficient for tumor resection. However, DTI tractography was needed for WMT concerning the lesion.
Maesawa et al. (2010) [27]	CST, Pyramidal tract	Preoperative DTI tractography depicted for surgical planning.	Intraoperative DTI tractography illustrated with conditions.	Craniotomy, Microsurgery	Yes, surgical planning needed to revise intraoperatively.	GTR: subtotal and greater in 85.7%, partially in four patients.	Intraoperative tractography gave a more accurate result than preoperative DTI tractography.
Bello et al. (2010) [20]	CST, inferior frontal-occipital fasciculus, Inferior longitudinal fasciculus, UNC, SLF	Preoperative DTI tractography was used for surgical approach.	DTI reconstruction was tested intraoperatively, combined with DES.	Craniotomy, Awake surgery	No	Postoperative deficits: neurological examination improved.	DTI tractography reconstruction corresponded with intraoperative subcortical mapping.
Buchmann et al. (2011) [32]	CST, Pyramidal tract	Preoperative DTI fiber tracking depicted for surgical planning.	DTI reconstruction was tested intraoperatively, combined with MEP.	Craniotomy	Yes, post hoc reviewed DTI images suggested changes in surgical approach, but only in one case	GTR: incomplete resection in seven patients. Postoperative deficits: temporary impairment after surgery, permanent in two patients.	DTI fiber tracking did not influence the surgical planning or the intraoperative course.
Hajiabadi et al. (2015) [21]	Optic Radiation, Visual pathway	Preoperative DTI depicted WMT fiber for surgical planning.	Intraoperative DTI revealed chiasm crossing fibers undetected by preoperative DTI.	Trans-sphenoidal sinus surgery, transcranial surgery.	No	Postoperative deficits: VIS significantly improved, except for one patient.	The intraoperative DTI finding predicted the visual outcome after tumor resection.
Faust and Vajkoczy (2016) [8]	Optic Radiation	Preoperative DTI tractography pre-determined fiber shift of OR.	NR	Temporal lobe surgery	Yes, pre-determined by pattern OR fiber shift.	GTR: total of 90% incomplete resection, 9% subtotal, and 1% partially removed. Postoperative deficits: VFD only 4%.	Surgical approaches were pre-determined by the pattern of OR fiber shifts depicted by DTI.
Zakaria et al. (2017) [28]	CST, Superior longitudinal fasciculus, and Arcuate Fascicles	Preoperative brain mapping, either for motor or language pathway was compared to non-mapping control.	NR	Craniotomy	No	Postoperative deficits: improved in the brain mapping group compared to the non-mapping group.	Automated whole-brain tractography mapping patients had more significant results in patients’ postoperative recovery.
Alexopoulos et al. (2019) [29]	Pyramidal tracts and superior thalamic radiations, SLF, IFOF, ILF, posterior thalamic radiations	Preoperative DTI tractography depicted for surgical approach by type of white matter tract characterization.	NR	Non-surgical	Yes, from total resection decision to subtotal	GTR: total resection in eight patients, and subtotal in seven patients.	DTI WM tractography identified WMT for better surgical outcome, but not operative approach.
Aibar-Duran et al. (2020) [23]	Pyramidal tract, inferior frontal-occipital fasciculus, Optic pathway, Inferior longitudinal fasciculus, aslant tract.	Preoperative DTI tractography was performed and compared to non-DTI control.	Evaluation of intraoperative navigated tractography on surgery time.	Awake surgery	No	GTR: more significant complete resection in DTI group compared to non -DTI group. Postoperative deficits: the development of new deficits was doubled in non- DTI group patients.	Intraoperative navigated tractography shortened the awake surgery time.
Zhang et al. (2020) [22]	Arcuate fascicles, pyramidal tract	Preoperative DTI tractography used for surgical approach.	NR	Craniotomy	No	GTR: no significant difference between DTI group compared to control group. Postoperative deficits: improved in trial group compared to control group, relating to KPS score.	The MRI scan was sufficient for tumor resection, and DTI tractography was needed for WMT evaluation concerning the tumor.
Xiao et al. (2021) [30]	CST	Preoperative DTI tractography was used for surgical approach.	DTI/DTT accuracy validated by DcCS	Craniotomy	Yes, surgical approaches changed based on the DTI finding.	Postoperative deficits: DTI prediction of postoperative deficits correlates to mRS score.	DTT is a valuable tool for surgical management of brainstem glioma.
Voets et al. (2021) [31]	CST, Arcuate, SLF, IFOF, Optic radiation, ILF.	Preoperative DTI tractography used for surgical approach.	Intraoperative subcortical stimulation was used	Awake surgery	No	Postoperative deficits: predictions of postoperative deficits were accurate and were preserved.	Preoperative DTI predictions were accurate in localization of tract, and postoperative DTI predicted recovery potential.

Abbreviation: NR; not recorded, DTI;diffusion tensor imaging, DTT; diffusion tensor tractography, MRI; magnetic resonance imaging, GTR; gross total resection, DES; direct electric stimulation, MEPS; motor evoked potential, HGG; High-grade gliomas, VIS; visual impairment score, VFD; visual field defects, SLF; superior longitudinal fasciculus, ILF; inferior longitudinal fasciculus, IFOF; inferior fronto-occipital fasciculus, UNC; uncinate fasciculus, CST; corticospinal tracts, OR, optic radiation, WMT; white matter tracts, WM; white matter, mRS; modified Rankin scale, KPS; Karnofsky performance score, mm; milimeter meter, DcCS; Direct subcortical stimulation.

## Data Availability

The data supporting reported results can be found in the main article or Appendix A.

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
