# Peer review of "The Utilization of Diffusion Tensor Imaging as an Image-Guided Tool in Brain Tumor Resection Surgery: A Systematic Review"

_cancers, 2022, doi:10.3390/cancers14102466_

Round 1

Reviewer 1 Report

The authors provide a systematic review on the impact of DTI on extent of resection and safety during brain surgery. The methodology is sound. However, the manuscript lacks various important aspects. Also, the manuscript would benefit from a thorough professional language editing.

I would recommend addressing the following aspects:

DTI just as fluorescence is simply tool among many and its impact on neurosurgical outcome depends on multiple factors. 

The authors should not mix up infiltrative (diffuse gliomas) and non-infiltrative (vascular malformations, most brain metastases) lesions in their analysis as biology of infiltration may affect fiber tracts differently

A discussion of limitations and technical aspects (FA, etc.) is missing

It is unclear to the reader what the authors mean by intraoperative DTI

“First of all, DTI is the only tool for visualisation of white matter 247 tract pathways to assure the anatomy and microstructure integrity of the tracts and the localising of 248 tumours concerning the fibre’s tracts [1,7-8,19-32].” – DTI is not able to assure integrity of tracts, intraoperative mapping and monitoring is the only intraoperative method to provide neurophysiological feedback

The conclusions last sentence should be deleted

Reviewer 2 Report

I appreciate this approach to provide a thorough overview. However, this article doesn't add any helpful information to the literature. For some reasons

1) the mindset that DTI itself could improve anything in surgery and outcome is 15 years old. Today we know better. There are so many studies investigating the accuracy of DTI via subcortical stimulation or lesions/deficits or ioMRI which should also be provided in order to give the reader what they need: trust or doubt if they can rely on the data 

2) pure DTI is a very old approach. Currently, there are a bunch of articles out dealing with function-based tractography. We know that white matter are fibers - of course - but only functional information by DCS, MEG or TMS are able to assign specific functions to tractography

3) there are also many other techniques of tractography rather than DTI; please provide

Round 2

Reviewer 1 Report

happy with the changes made